# Attempt for a Recombinant Thrombomodulin Alpha Treatment in a Rat Disseminated Intravascular Coagulation Model Using Yamakagashi (*Rhabdophis tigrinus*) Venom

**DOI:** 10.3390/toxins14050322

**Published:** 2022-05-02

**Authors:** Akihiko Yamamoto, Takashi Ito, Toru Hifumi

**Affiliations:** 1Management Department of Biosafety and Laboratory Animal, and Pathogen Bank, National Institute of Infectious Diseases, Tokyo 208-0011, Japan; 2Department of Biomedical Laboratory Sciences, Faculty of Life Sciences Kumamoto University, Kumamoto 860-8556, Japan; tito@kumamoto-u.ac.jp; 3Emergency and Critical Care Medicine, St. Luke’s International Hospital, Tokyo 104-8560, Japan; hifumitoru@gmail.com

**Keywords:** recombinant thrombomodulin alpha, Yamakagashi (*Rhabdophis tigrinus*) venom, rat disseminated intravascular coagulation (DIC) model

## Abstract

Yamakagashi (*Rhabdophis tigrinus*) inhabits Japan widely, and incidents involving its bites occur every year. Its bite causes disseminated intravascular coagulation when the amount of infused venom is high, and it can be fatal if treatment with Yamakagashi antivenom is delayed. Although Yamakagashi antivenom is used for treating Yamakagashi bites, it is an unapproved drug and its capacity for storage is limited. Hence, it is difficult to administer to patients promptly. As a therapeutic agent for this bite, we investigated the application of recombinant thrombomodulin alpha, a commercially available disseminated intravascular coagulation therapeutic agent. Its therapeutic effect on Yamakagashi venom was confirmed in a coagulation system of human plasma using in vitro Yamakagashi venom as well as a rat experimental model of disseminated intravascular coagulation using in vivo Yamakagashi venom. The administration of recombinant thrombomodulin alpha induced an effect that prolonged the blood coagulation time of Yamakagashi venom in vitro, and the drug was administered in vivo within 0.5 h after the administration of Yamakagashi venom to save rats. Blood coagulation markers such as platelet count, prothrombin time, fibrinogen concentration, and D-dimer levels recovered to normal values in rats. Therefore, recombinant thrombomodulin alpha may be used as a therapeutic agent for Yamakagashi bites.

## 1. Introduction

Disseminated intravascular coagulation (DIC), a severe clinical condition caused by an underlying disease, involves a markedly continuous and widespread activation of coagulation in the circulating blood and the formation of numerous microvascular thrombi [1,2]. DIC is also caused by some snakebites [3].

Yamakagashi (*Rhabdophis tigrinus*), belonging to the family Colubridae, is a rear-fanged venomous snake. Its fangs are short with no groove. This venomous snake is widespread in East Asia. In Japan, it is commonly found in the paddy fields, and it feeds primarily on frogs [4]. A severe case of its bite is rare because the venom can be introduced into the skin of the person only when it attacks with the rearmost fangs. However, several cases of bites requiring treatment are reported every year [5]. Regarding the chemical characterization of *R. tigrinus* venom, the higher molecular mass fraction of the venom contains a prothrombin activator. Both the prothrombin time (PT) and activated partial thromboplastin time of human plasma were found to be shortened by the addition of this snake venom. Thrombin formation was determined using SDS-PAGE and chromogenic substrates. The venom fractions also exhibited specific proteinase activity on human fibrinogen (FIB), but the substrates for matrix metalloproteinase, such as collagen and laminin, were not hydrolyzed [6]. In an in vivo experiment conducted using mice, not only was local hemorrhage observed, but also systemic subcutaneous, pulmonary, and subendocardial hemorrhages as well. Microthrombi in the alveolar capillaries and glomerulus were also observed [7]. In cases of bites, systemic hemorrhage, DIC, and acute renal failure have been observed [8,9], and two fatal cases of acute pulmonary edema and cerebral hemorrhage have also been reported [10,11].

Currently, an unapproved drug, *R. tigrinus* antivenom, is used as a therapeutic agent for *R. tigrinus* envenomation, but there are restrictions on its use. *R. tigrinus* antivenom is a horse plasma-derived preparation immunized with *R**. tigrinus* venom, and when administered to humans, there is a risk of causing adverse reactions such as serum sickness and anaphylaxis due to animal protein. There are also several approved drugs already on the market for the treatment of DIC [12,13,14]. Among those drugs there is no record of DIC treatment with venomous snakes, but recombinant thrombomodulin alpha (rTM) is often used as a drug that is always available in clinical practice [15]. None of the approved DIC treatments listed here have been used to treat snake-venom-derived DIC. Therefore, in the future, we plan to investigate the therapeutic effect of snake venom on DIC. Thrombomodulin is a glycoprotein present on vascular endothelial cells, and it was revealed in 1982 by Dr. Esmon et al. from the University of Oklahoma, USA, as a physiological anticoagulation factor responsible for regulating blood coagulation in the body [16]. The gene encoding thrombomodulin was isolated, and it was clarified that its active site exists in the extracellular domain. Moreover, only the extracellular domain in animal cells was successfully produced by genetic engineering. A clinical trial of this soluble human thrombomodulin was conducted, and a manufacturing and marketing approval application was submitted for the treatment of DIC in 2006, following which the drug was approved in January 2008 [17,18]. Studies have reported that rTM exerts a therapeutic effect on DIC caused by infectious diseases, malignant tumors, trauma, and gynecological diseases [19,20,21]. The mechanism underlying the action of rTM involves promoting the activation of protein C by thrombin, and the generated activated protein C decomposes the coagulation-promoting factors Va and VIIIa using protein S as a coagulation factor to generate thrombin [22]. Through this process, rTM suppresses the blood coagulation reaction.

In this study, we confirmed the effect of rTM on *R. tigrinus* venom in an in vitro blood coagulation system as a putative therapeutic agent for the DIC state of *R. tigrinus* bites. We also investigated the therapeutic effect of rTM in an in vivo rat DIC model.

## 2. Results

### 2.1. Effect of rTM on the In Vitro Human Blood Coagulation Activity of R. tigrinus Venom

To confirm the effect of rTM on *R. tigrinus* venom, we conducted an experiment in an in vitro blood coagulation measurement system using human standard plasma. We examined how the coagulation effect of different concentrations of *R. tigrinus* venom affected human standard plasma with the simultaneous addition of a constant concentration of rTM (Figure 1).

When a variable amount of *R. tigrinus* venom was added to human standard plasma and heated, as shown by the black circle in Figure 1, the blood coagulation time was shortened depending on the concentration of the added venom. The addition of 10 μg of rTM to this measurement system significantly extended the coagulation time due to the addition of *R. tigrinus* venom, as indicated by the black square in Figure 1.

A statistical analysis of the values of the coagulation time of *R. tigrinus* venom alone or with different additions to the in vitro human plasma and the addition of 10 μg (67 U) of rTM to the venom revealed that both regressions were established with a risk rate of 1%. In addition, the reactions under the two experimental conditions also showed a significant difference at a risk rate of 1%. Furthermore, when these two experimental conditions were compared by the parallel line assay method, it was found that the linearity and concurrency of each regression curve cannot be denied. In addition, the coagulation activity of *R. tigrinus* venom on human plasma was significantly suppressed by the addition of rTM. The degree of suppression with the addition of rTM was calculated to be 13.85% with *R.tigrinus* venom alone, and the 95% confidence limit was calculated to be 18.5% for the upper limit and 10.73% for the lower limit.

Addition of 10 ug of rTM to *R. tigrinus* venom suppressed the coagulation activity of the venom up to 13.85%. From this, it was calculated to be 2.1 mg when calculating how much the amount of *R. tigrinus* venom was suppressed by the addition of 1 mg of rTM.

### 2.2. Life-Saving Effect of rTM on the Rat DIC Model by R. tigrinus Venom

In the in vitro measurement system shown in Figure 1, the coagulation activities of the *R. tigrinus* venom were inhibited by rTM because other activities cannot have been inhibited. Therefore, a confirmation experiment was conducted using the in vivo measurement system of rTM. We used a previously described rat DIC model as a system for measuring the activity of *R. tigrinus* venom in vivo [23]. Briefly, 1 mg/kg of rTM was administered intravenously to this measurement system at 2, 0.5, 0.33, and 0.17 h after venom administration. As shown by the results depicted in Figure 2, the administration of 1 mg/kg of rTM 2 h after venom administration did not improve the survival time of the rats. However, the administration of rTM 0.5, 0.33, and 0.17 h after venom administration saved 33.3%, 100%, and 66.7% of the rats, respectively. When the significant difference between the two groups was calculated using a Kaplan–Meier analysis for the life-saving effect shown in Figure 2 between the untreated group of the rat DIC model with *R. tigrinus* venom and the rTM-administered group with different administration times, rTM was administered 0.33 h later. In the group in which 100% of the rats were saved, the difference was significant with a 5% risk.

### 2.3. Therapeutic Effect of rTM on the Rat DIC Model with R. tigrinus Venom: Effect on Four Blood Coagulation Marker Values

On the basis of the life-saving effect of rTM on the rat DIC model shown in Figure 2, the changes in platelet count, PT, FIB concentration, and D-dimer values were investigated as blood coagulation markers for the rats used in the experiment. The results are illustrated in Figure 3, which shows the time course of the blood coagulation markers in rat blood treated with rTM after 2, 0.5, 0.33, and 0.17 h without treatment and after *R. tigrinus* venom administration to the rat DIC model treated with *R. tigrinus* venom. In this figure, A shows the platelet count, B shows PT, C shows the FIB concentration, and D shows the change in each blood coagulation marker with the passage of time of the D-dimer concentration.

From the results shown in Figure 3, in the rat DIC model to which only *R. tigrinus* venom was administered, the platelet count peaked at 8 h after administration of the venom, the PT time was delayed to the measurement limit of 120 s after 2 h, the FIB concentration decreased to the measurement limit after 2 h, and D-dimer was observed to have a transient increase peaking at 2 h. In comparison, two hours later, the fate of the blood coagulation factors was observed in the rTM-administered group, which was not significantly different from that in the venom-administered group. On the other hand, in the rTM 0.5, 0.33, and 0.17 h post-administration groups, recovery of platelet count, PT, and FIB levels was observed, and D-dimer was scarcely observed. A statistical comparison of therapeutic effects in rat DIC models was made between the untreated group and the rTM 2, 0.5, 0.33, and 0.17 h post-treatment groups. The platelet count, PT, and FIB concentrations were found to be significant at a 1% risk rate in the group receiving rTM 0.33 and 0.17 h after venom administration compared to the untreated group (Indicated by ** in Figure 3). A comparison of the D-dimer concentrations calculated that the group receiving rTM 0.33 h and 0.17 h later, including the group receiving rTM 0.5 h after venom administration, was significant at a risk rate of 1%.

## 3. Discussion

In this study, we confirmed the action of rTM on *R. tigrinus* venom using two methods, namely, in vitro and in vivo. First, in the in vitro blood coagulation system using standard human plasma, the administration of rTM canceled the blood coagulation activity of 2.1 mg of *R. tigrinus* venom per 1 mg. Second, the in vivo method saved 33.3%, 100%, and 66.7% of the rats after the administration of 1 mg/kg of rTM 0.5, 0.33, and 0.17 h after venom administration in the rat DIC model with *R. tigrinus* venom, respectively. The rat survivors recovered from thrombocytopenia, prolongation of PT, and decreased FIB concentrations, which represent the changes in blood coagulation markers that occur in the rat DIC model, with a slight transient increase in D-dimer levels.

As the in vitro blood coagulation system consisted only of human standard plasma, calcium chloride, and *R. tigrinus* venom, rTM may act directly on *R. tigrinus* venom to suppress its blood coagulation effect. However, coagulation occurs even when calcium chloride is added to human standard plasma without the addition of *R. tigrinus* venom, which was performed as a control experiment, and was heated at 37 °C. When rTM was added to this reaction, a concentration-dependent prolongation effect of the added rTM was observed on the coagulation time. This clearly showed that rTM does not act directly on *R. tigrinus* venom. Moreover, as no experiments have been conducted with the addition of rTM inhibitors, the mechanism by which the activity of *R. tigrinus* venom is canceled has not yet been elucidated in the in vitro experiments that were conducted in this study. Comparing the cancelling effects of *R tigrinus* antivenom on the same measurement system, 1 mg of rTM is equivalent to 60% of one vial of *R. tigrinus* antivenom [24].

Next, the effect of rTM on the rat DIC model treated with *R. tigrinus* venom showed that it could partially or completely save the rat with the administration of 1 mg/kg of rTM within 0.5 h of the venom administration (Figure 2). This observation reflects the above-described in vitro experimental results (Figure 1). However, even when the same amount of rTM was administered 2 h after the administration of venom, no life-saving effect was observed (Figure 2 and Figure 3). Considering the relationship between the administration time and life-saving effect, the administered venom is carried systemically by the bloodstream, and the rat is saved by the administration of rTM within 0.5 h after the administration of venom before exerting its effect. Nevertheless, according to the in vitro experimental results, it is unlikely that the administered rTM will have a direct effect on the toxic neutralization of *R. tigrinus* venom. Compared with the effect of *R. tigrinus* antivenom in the in vivo experimental system, the life-saving effect on rats was similar. In addition, a comparison of the changes in blood coagulation markers in life-saving rats that recovered from thrombocytopenia, prolongation of PT, and decreased FIB concentrations was observed [23]. The difference in the effects of rTM and *R. tigrinus* antivenom in the in vivo experimental system appeared in the amount of D-dimer as a blood coagulation marker. Antivenom had little effect on the appearance of D-dimer, but rTM significantly suppressed the appearance of D-dimer [23]. D-dimer is normally undetectable or only detectable at a very low level unless the animal body is forming and breaking down significant blood clots. A positive or elevated D-dimer test result may indicate that the animal has a blood clotting condition, but it doesn’t guarantee that they have one [24]. In the rat DIC model, D-dimer increased, and coagulation occurred systemically, but the fact that D-dimer was hardly detected within 0.5 h of administration of rTM (Figure 3D) indicates that coagulation was suppressed. It was speculated that the difference between the two was due to the difference in the mechanism of action.

rTM is a glycoprotein present on vascular endothelial cells and acts as a physiological anticoagulation factor responsible for regulating blood coagulation in the body [16,17,18]. rTM has been used to treat DIC caused by infectious diseases, cancer, and sepsis [19,20,21]. Regarding the possibility of treating DIC caused by snake venom, it was found that the administration of *R. tigrinus* venom to the rat DIC model could save the lives of rats. A horse-specific antibody against the venom has been used for bites caused by *R. tigrinus*. However, owing to the prolonged transport time of this antibody, it is difficult to administer it to patients with rapid bites. Therefore, as rTM is always available in all medical institutions as an approved drug for DIC treatment, it is easy to administer it to patients bitten by *R. tigrinus*. Our experimental results showed that the toxic activity could not be canceled unless rTM was administered to the bite patient in a short time after being bitten by *R. tigrinus*. Therefore, rTM can be considered a potential therapeutic agent for bites caused by *R. tigrinus.*

The mechanism underlying the action of rTM involves promoting the activation of protein C by thrombin, and the generated activated protein C uses protein S as a coagulation factor to decompose the coagulation promoters Va and VIIIa to produce thrombin [22]. Through this process, rTM suppresses the blood coagulation reaction. The mechanism underlying the life-saving effect of rTM in the rat DIC model treated with *R. tigrinus* venom was revealed in the in vitro and in vivo experiments using inhibitors for each reaction of the anticoagulant as described earlier.

From the results of the in vitro and in vivo experiments conducted in this study, rTM did not directly neutralize *R*. *tigrinus*, unlike antivenom, and the survival of the rats was within 0.5 h after administration of the venom. Therefore, considering future clinical use, it is necessary to consider: (1) dealing with the toxicity of *R*. *tigrinus* venom other than DIC, such as hemolytic activity, for patients with *R*. *tigrinus* bites; and (2) it has the characteristic that it is required to be administered as soon as possible. In addition, bleeding and damage to the liver and bile duct system have been reported as side effects of high doses of rTM, so this point must also be taken into consideration.

Regarding the limitations of this study, although we demonstrated the results of in vitro and in vivo experiments, the number of experiments in each model was as small as three. Moreover, because only one concentration of rTM was used, the conclusions drawn from the experimental results are limited. In the future, the effect of rTM on *R. tigrinus* venom should be further clarified by conducting studies with consideration of the rTM concentration, frequency of administration, and administration time even for in vivo experiments.

## 4. Conclusions

As a putative agent for the DIC state of *R. tigrinus* bite, we confirmed the effect of rTM on *R. tigrinus* venom in the in vitro blood coagulation system, and our results regarding the therapeutic effect of rTM on the in vivo rat DIC model showed that the administration of rTM in a short time after venom administration could save the life of the rat.

## 5. Materials and Methods

### 5.1. Blood Coagulation Assay

The *R. tigrinus* venom solutions were prepared with different concentrations in Owren’s buffer (Helena Biosciences Europe; London, UK). For determining the coagulation time due to *R. tigrinus* venom, 0.05 mL of venom and 0.05 mL of CaCl2(25 mM) were mixed at 37 °C and 0.05 mL of standard human plasma (98406012: Sysmex Co., Kobe, Japan) was added to the semi-automatic blood coagulation measuring device (CA-101: Sysmex Co., Kobe, Japan). The human plasma used in this experiment was produced from healthy human blood that was negative for HIV, Hepatitis B, and C virus; was treated with citric acid; stabilized with HEPRS buffer (12 g/L); and then lyophilized. When dissolved in 1 mL of distilled water for injection, the components are blood coagulation factors II, V, VII, VIII, IX, X, XI, XII, and XIII, as well as protein C, antithrombin III, C1 inhibitor, and VWF. This is included in the concentration of 0.87 to 1.17 IU/mL. In addition, the fibrinogen concentration is 2.49 mg/mL. The rTM used in this experiment has an activity of 6700 U/mg. Next, to confirm the effect of rTM, 0.025 mL of venom and 0.025 mL of rTM were added, 0.05 mL of CaCl2 was mixed at 37 °C, and 0.05 mL of human plasma was added to this solution. The CA-101 device senses the addition of human plasma, heats it at 37 °C while stirring the contents with a rotor in the cuvette, and measures the time until the purification of the coagulated mass. The comparison between the coagulation activity of *R**. tigrinus* venom and the suppression of its activity by rTM was performed using the parallel line assay method.

### 5.2. Animal Preparation

Male Sprague Dawley rats (JAPAN SLC, Inc., Shizuoka, Japan) aged 12 weeks were housed in separate cages in a temperature-controlled room under a 12 h/12 h light/dark cycle. They were fed on a standard laboratory diet and given water *ad libitum*. The weight of the rats used in the experiment ranged from 300 to 350 g. All surgical and experimental procedures were approved by the Animal Care and Use Committee and conformed to the Guidelines for Animal Experimentation (approval number: 118065, approval date: 13 December 2018).

Under general anesthesia using 4% isoflurane induction, a polyethylene catheter (PE-60) was inserted into the femoral artery of the rat for bleeding. Another catheter (PE-50) was inserted into the femoral vein for the administration of saline solution and drugs. All catheters were filled with heparinized saline (100 U/mL), placed under the skin, and then opened under the neck to prevent the catheter from coming off when the animal awakened. After placing these two catheters, the rats were subjected to the experiment after a 48-h post-treatment recovery period. Blood sampling and drug administration to the rats were performed after anesthesia induction with 4% isoflurane.

### 5.3. Snake Venom

*R. tigrinus* venom was extracted from the Duvernoy’s gland. Toxic glands were collected from 100 male and female *R. tigrinus* measuring ≥80 cm in body length from throughout Japan. The glands were excised, cut into small pieces, and centrifuged with distilled water, after which the supernatant was lyophilized [25]. The venom powder was dissolved in distilled water and centrifuged again to remove the mucous substance that was contained during venom extraction. The lyophilized venom was stored in a refrigerator.

The intravenous LD50 value of the venom was 5.3 μg/20 g mouse [10].

### 5.4. Experimental Protocols

The rats were initially randomly divided into five groups to evaluate lethality as follows: (a) 300 μg of *R. tigrinus* venom was intramuscularly administered; 1 mg/kg of rTM was administered (b) 2, (c) 0.5, (d) 0.33, and (e) 0.17 h after *R. tigrinus* venom (300 μg) administration (each time: *n* = 3, respectively). Blood samples were collected at 0, 2, 4, 8, 24, 48, 72, 96, and 120 h after the start of the experiment (1500 μL of blood was collected at each time point). An equal volume of saline was administered through the femoral vein after bleeding. Blood samples were anticoagulated with sodium citrate and centrifuged immediately at 3000× *g* for 10 min, and the plasma supernatant was separated.

### 5.5. Test Principle in the Measurement of Platelet Count

Platelet counts were evaluated using an automated system on ADVIA 2120i (Siemens Diagnostic Solutions, Milan, Italy). ADVIA counts platelets by flow cytometry based on the principle of light scattering. The platelets are identified by their size (<30 FL, low-angle light scatter) and refractive index (*n* = 1.35–1.40 or high-angle light scatter). After blood was collected from the rats, EDTA was added to whole blood to prevent coagulation. The cells were stored at 4 °C until the end of the experiment, and after the listed experiments were completed, the platelet count was measured with ADVIA 2120i.

### 5.6. Test Principle in the Measurement of PT

An individual plasma sample was placed in a measuring test tube of CA-50 AutoAnalyzer (Sysmex Co., Kobe, Japan). The coagulation reaction detection method irradiates red light (660 nm) onto a mixture of blood plasma and reagent, detects the change in turbidity (when fibrin clots are formed) as the change in scattered light, and measures the coagulation time (s). It detects within the maximum detection time and measures the result. The typical maximum detection time is 120 s for PT. The coefficient of variation (CV) of the PT measurement is ≤2%. The data for CV are variation coefficients for the coagulation times of the change in activity (%) obtained from ten analyses of Dade Behring Ci-Trol Level 1 (control plasma) with a PT reagent.

### 5.7. Test Principle in the Measurement of FIB Concentration

An individual plasma sample diluted one-tenth with specific reagents was placed in a measuring test tube of CA-50 AutoAnalyzer (Sysmex Co., Kobe, Japan). The coagulation reaction detection method irradiates red light (660 nm) onto a mixture of blood plasma and reagent, detects the change in turbidity (when fibrin clots are formed) as the change in scattered light, and measures the coagulation time (s). The range of analysis for the FIB concentration can be from 50 to 450 mg/dL. The CV of FIB measurement is ≤4%. The data for CV are variation coefficients of the coagulation times of the change in activity (%) taken from ten analyses of Dade Behring Ci-Trol Level 1 (control plasma) with Dade Behring Fibrinogen Determination Reagents.

### 5.8. Test Principle in the Measurement of D-Dimer Levels

The individual plasma levels of D-dimer were measured using a latex photometric immunoassay (LPIA)-NV7 with an LPIA ACE D-Dimer II Kit (LSI Medience, Tokyo, Japan), as reported previously [26]. Briefly, 4 μL of plasma was dispensed with 144 μL of R-1 solution into the reaction cuvette. After a 2-min stabilizing period at 37 °C, 48 μL of R-2 suspension was dispensed, which contained latex particles coated with the anti-D-dimer antibody JIF-23. The increase in turbidity was evaluated during the 7-min reaction period at 37 °C. The analysis range for D-dimer levels was 0.5–48 μg/mL, and samples beyond this range were remeasured with dilution. The within-run CV was <10%. The cross-reactivity to rat D-dimer was confirmed in the preliminary test using rat plasma samples treated with calcium ions and different concentrations of tissue plasminogen activator.

### 5.9. Statistical Analysis

The uniformity of data distribution in each group was assessed using Bart-lett’s test. As a result, when comparing data groups with confirmed homogeneity, we used a bidirectional analysis method to compare the groups. We also used the Mann–Whitney U test to compare data groups with negative homogeneity. In the in vitro blood coagulation system comparison, the parallel line assay method was used. The survival study used Kaplan–Meier analysis to analyze the data. In addition, the binary placement method was used to compare the blood coagulation markers. Statistical analyses were conducted using the JMP version 11 software (SAS Institute, Cary, NC, USA). A two-sided probability value of <0.05 was considered statistically significant in all analyses.

## Figures and Tables

**Figure 1 toxins-14-00322-f001:**
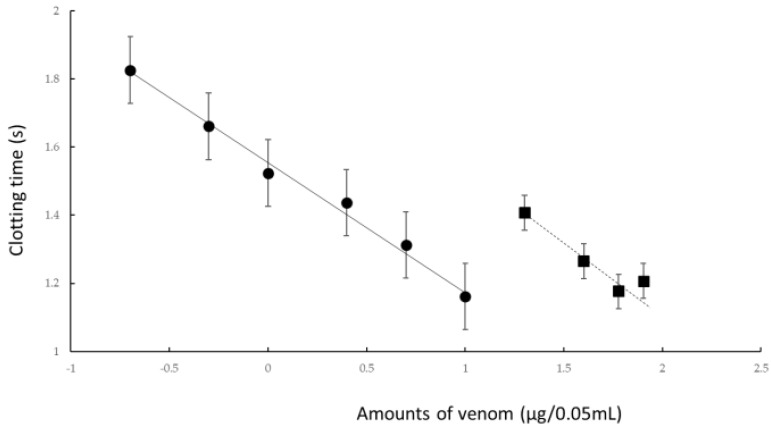
Effect of rTM on the in vitro human blood coagulation activity of *R. tigrinus* venom. Black circles indicate the coagulation time of standard plasma by *R. tigrinus* venom at different concentrations (●). Black squares show the effect of prolonging the coagulation time of the venom in the presence of 10 μg of rTM (█). The *x*-axis represents the logarithmic value of venom concentration and the *y*-axis represents the logarithmic value of plasma coagulation time. All values indicate the average ± SD of three experiments.

**Figure 2 toxins-14-00322-f002:**
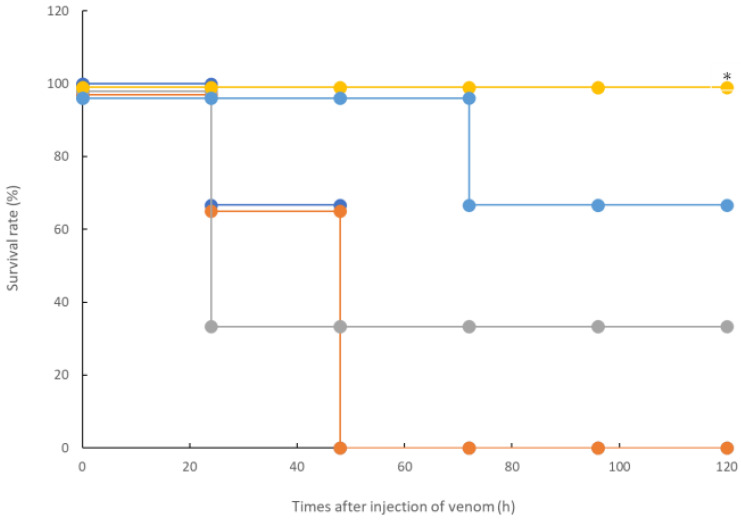
Life-saving effect of rTM against lethal doses of *R. tigrinus* venom. Orange circles indicate the survival time after the intramuscular administration of 300 μg of venom (●). The survival time when rTM was administered by varying the time after venom administration is shown. It is represented as blue circles after 2 h (●), light blue circles after 0.5 h (●), yellow circles after 0.33 h (●), and grey circles after 0.17 h (●). Each group consists of three rats. *: indicates statistically significant (*p* < 0.05).

**Figure 3 toxins-14-00322-f003:**
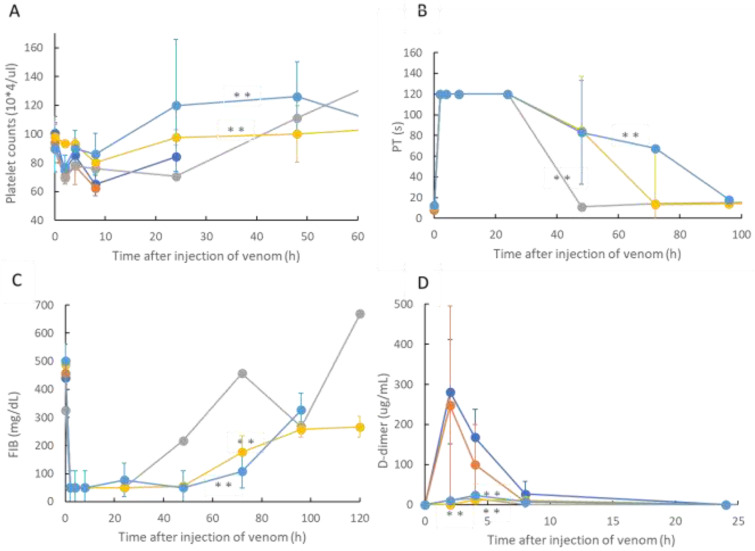
Changes in platelet count, PT, FIB concentration, and D-dimer levels over time in rats after the intramuscular administration of 300 μg of *R. tigrinus* venom. In this figure, (panel **A**) shows the change in platelet count over time, (panel **B**) shows PT, (panel **C**) shows FIB concentration, and (panel **D**) shows the time change of D-dimer levels. In each panel, orange circles (●) indicate changes over time in rats after venom administration, blue circles (●) indicate changes with rTM administration 2 h later, light blue circles (●) indicate changes at 0.5 h after venom administration, yellow circles (●) indicate changes at 0.33 h after venom administration, and grey circles (●) indicate changes with rTM administration 0.17 h later. The time course of the rat is shown. Each group consisted of three rats. The value displayed is average ± SD. **: indicates statistically significant (*p* = 0.01).

## Data Availability

Data are available in a publicly accessible repository that does not issue DOIs.

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
