# Peer review of "Attempt for a Recombinant Thrombomodulin Alpha Treatment in a Rat Disseminated Intravascular Coagulation Model Using Yamakagashi (Rhabdophis tigrinus) Venom"

_toxins, 2022, doi:10.3390/toxins14050322_

Round 1
Reviewer 1 Report
The potential use of the recombinant drug thrombomodulin alfa as an anticoagulant against the main effect of envenomation caused by the Yamakagashi snake deserves preliminary studies in in vitro and in vivo models, as presented in the present study.
But I would like to suggest some corrections and improvements in the present manuscript:
- An interesting question is whether fibrinolytic drugs could have a potential therapeutic effect in the treatment of disseminated intravascular coagulation caused by such a venom;
-Describe more adequately the results described in item 2.3. To say that something "increased" or "decreased" is not technically adequate, mentioning data and statistics;
-Correct the respective shape of all subtitles that mention "black square";
-Improve the representation of the dashed lines of the graphs in Figure 3;
-D-Dimer is not well explained as a marker of anticoagulation;
- It was not clear why animals treated with thrombomodulin alfa 0.33 h after venom injection was not as effective in 100% survival of animals, when compared to those who received after 0.5 h, in which I had 100% survival. ;
-Based on the known procoagulant mechanism of action of the Yamakagashi snake venom, the authors could propose a mechanism of action of the recombinant thrombomodulin alpha, associating the results of increased survival and four blood coagulation markers analyzed. Part of the discussion was description of results;
- The calculation of the dose of venom administered to rats, in the in vivo protocol, is not very clear, since it was based on LD50 data in mice. What is the average weight of the animals used in this protocol?
-For the extraction of venom by the glands of Yamakagashi snakes, as described in item 5.3, did you not need approval from an ethics committee?
- Item 5.4 does not make sense as a description of an experimental protocol, and seems to be a justification based on the results obtained. I suggest removing this item.
-Describe in more detail the experimental protocol referring to item 5.6 on platelet count. Was it done in whole blood?
Author Response
Dear reviewer 1,
Apr. 29st, 2022
Thank you for your comment on the draft of Toxins-1679860.
Below is a response to each of the three reviewers who received many useful comments. 
We will attach the toxins-1679860 (2) .pdf file as a revised paper.
To reviewer 1.
The potential use of the recombinant drug thrombomodulin alfa as an anticoagulant against the main effect of envenomation caused by the Yamakagashi snake deserves preliminary studies in in vitro and in vivo models, as presented in the present study.
But I would like to suggest some corrections and improvements in the present manuscript:
We responded to each of the 11 specific questions as follows.
- An interesting question is whether fibrinolytic drugs could have a potential therapeutic effect in the treatment of disseminated intravascular coagulation caused by such a venom;
Until now, antivenom has been mainly used for the treatment of DIC due to snake venom, but this paper first investigated the potential therapeutic effect of general anti-DIC drugs.
- -Describe more adequately the results described in item 2.3. To say that something "increased" or "decreased" is not technically adequate, mentioning data and statistics;
As pointed out, the description of the data in Figure 3 shown in Result 2.3 was reviewed and rewritten into a representation with statistical calculations.
- Correct the respective shape of all subtitles that mention "black square";
As you pointed out, the groups displayed in Figures 2 and 3 including the black square display are difficult to understand, so we changed the display of these two tools to color display.
- Improve the representation of the dashed lines of the graphs in Figure 3;
As you pointed out, the group containing the dotted line displayed in Figure 3 is confusing, so we changed the display in Figure 3 to color.
- D-Dimer is not well explained as a marker of anticoagulation;
In response to the indication, we added a description about D-dimer to the discussion.
- It was not clear why animals treated with thrombomodulin alfa 0.33 h after venom injection was not as effective in 100% survival of animals, when compared to those who received after 0.5 h, in which I had 100% survival.;
Regarding the point you pointed out, we also do not understand the relationship between survival rate and administration time. The data used in this paper is based on 3 rats per group, so it cannot be denied that the power of detection is low.
- Based on the known procoagulant mechanism of action of the Yamakagashi snake venom, the authors could propose a mechanism of action of the recombinant thrombomodulin alpha, associating the results of increased survival and four blood coagulation markers analyzed. Part of the discussion was description of results;
Considering the points pointed out and the effect of rTM that suppresses D-dimer shown in Figure 3D, the inhibitory effect of R. tigrinus venom this time is presumed to be the suppression of thrombin by the activation of protein C by rTM. This point is mentioned in the discussion section.
- The calculation of the dose of venom administered to rats, in the in vivo protocol, is not very clear, since it was based on LD50 data in mice. What is the average weight of the animals used in this protocol?
Unfortunately, the LD50 data on the toxicity of the R. tigrinus venom used in this experiment are only for mice. However, in a rat DIC model paper (Reference No. 23) submitted to Toxins, it was stated that intravenous administration of 150 ug of R. tigrinus venom to rats resulted in immediate death.
The weight of the rats used in this study was 300g-350g.  The weight of the rat used in the experiment is additionally described in 5.2.
- For the extraction of venom by the glands of Yamakagashi snakes, as described in item 5.3, did you not need approval from an ethics committee?
The ethical approval of the method of collecting venoms from the snake you pointed out has been comprehensively approved by the Animal Care and Use Committee of the National Institute of Infectious Diseases.
- Item 5.4 does not make sense as a description of an experimental protocol, and seems to be a justification based on the results obtained. I suggest removing this item.
As you pointed out, the item of 5.4 has been deleted.
- Describe in more detail the experimental protocol referring to item 5.6 on platelet count. Was it done in whole blood?
For the measurement of platelet count, whole rat blood is added with EDTA to prevent coagulation and used as a sample. We added this point to the method.

Reviewer 2 Report
TOXINS Reference manuscript: Attempt of recombinant thrombomodulin alfa treatment to a rat disseminated intravascular coagulation model using
Yamakagashi (Rhabdophis tigrinus) venom. Submitted April 2022.
General Comments
In this manuscript the authors are describing their experimental data on the use of the recombinant protein Alpha Thrombomodulin, as a possible drug in the treatment of accidents with the Asian snake Rhabdophis tigrinus, seeking to inhibit complications of Disseminated Intravascular Coagulation. The authors studied the protective effects of treatment with Alpha Thrombomodulin in rats exposed to the venom experimentally and treated with recombinant Alpha Thrombomodulin after this exposure. Throughout the text, the authors report their findings with emphasis on the normalization of coagulation parameters in the blood of animals and from there they report the possibility of using this recombinant protein in the treatment of injured patients. The text is well-written, easy to understand, and theoretically well-founded. However, there are suggestions that authors can incorporate into a revised version to increase reader interest and improve some definitions.
Comments
1- Please, as written on line 38, substitute throughout of the text the words molecular weight for molecular mass. I know that several published texts bring this term, but it is incorrect! Molecules have no weight! Rice, meat have weight... Molecules have molecular masses. We don't have weight spectrometry but instead mass spectrometry....
2- Between lines 42 to 44 the authors wrote.... The venom fractions also exhibited specific proteinase activity on human fibrinogen (FIB), but the substrates for matrix metalloproteinase, such as collagen and laminin, were not hydrolyzed [6]. I ask … are there data on the hydrolysis of plasma and vascular extracellular matrix proteins such as Fibronectin, von Willebrand Factor and Vitronectin, which are Extracellular Matrix molecules involved in platelet adhesion and aggregation?
3- Between lines 50 and 51 the authors wrote....Currently, an unapproved drug, R. tigrinus antivenom, is used as a therapeutic agent for R. tigrinus envenomation, but there are restrictions on its use… What would these restrictions be, in addition to those already mentioned in the difficulty of access and non-approved drugs. Would the serum be a causative agent of severe allergenic responses? It would be good for the authors to discuss more details here about this! Indicating possible advantages of the proposed alternative!
4- Between lines 51 and 55 the authors wrote....There are also several approved drugs already on the market for the treatment of DIC [12–14]. Among those drugs, there is no record of DIC treatment with venomous snakes, but recombinant alpha thrombomodulin (rTM) is often used as a drug that is always available in clinical practice[15]. …..Although the authors have not carried out comparative analyzes between drugs already used in the treatment of DIC and recombinant alpha thrombomodulin, here, it would be interesting for them to comment on possible advantages of the proposed method and, in the future, make these comparisons in pre-clinical or clinical trials!
5- Between lines 70 and 71 the authors wrote....In this study, we confirmed the effect of rTM on R. tigrinus venom in an in vitro blood coagulation system as a therapeutic agent for the DIC state of R. tigrinus bites. ….It would be better to write putative therapeutic agent since the method is being tested and its initial pre-clinical phase!
6- Although Figure 1 shows robust data on the inhibition of blood clotting induced by R. tigrinus venom at different concentrations, caused by the presence of alpha thrombomodulin in the in vitro system using reference plasma. I missed more direct graphs of Prothrombin Times, Partial Activated Prothrombin Time, Thrombin Time, Platelet Count, Platelet Aggregation, Clotting Time, Clot Retraction, among other laboratory parameters used for blood clotting analysis!
7- Between lines 100 and 101 the authors wrote …In the in vitro measurement system shown in Figure 1, the activity of R. tigrinus venom was canceled by rTM. In my opinion, in a revised text It would be better to write that the coagulation activities of the venom were inhibited, because others activities cannot have been inhibited.
8- What does this horizontal line in figure 1 mean in the value of 1.3 cloting time? There are no comments in the figure legend!
9- Between lines 107 and 108, discussing about figure 2, the authors wrote …However, the administration of rTM 0.5, 0.33, and 0.17 h after venom administration saved 33.3%, 100%, and 66.7% of the rats, respectively. How do the authors explain that for a shorter time of administration of alpha thrombomodulin (0.17h), the protection was inferior to the time of 0.33h? Would it be individual variation of the animals? Also because only 3 animals were used and this number of animals could be insufficient for such an analysis. Sounds like a small population to me to assess an important postulated clinical effect!
10- Between lines 108 to 113, discussing about figure 2, the authors wrote …When the significant difference between the two groups was calculated using a statistical method for the life-saving effect shown in Figure 2 between … In my opinion, the authors must indicate the statistical method used
11- On the graphs shown in Figure 3, there is a lot of overlap, making it difficult to interpret the results. Did the authors not think of presenting the data in the form of histograms, which could make interpretation easier? Alternatively they could do the graph with different colors which would make the data interpretation easier too!
12- The authors already discuss in the abstract that although serum therapy is an efficient treatment for victims of Rhabdophis tigrinus snake bites, this treatment is difficult to use due to the difficulty in obtaining the serum and also because the serum is not officially approved by the regulatory agencies in Japan. Would it be interesting for them to show which treatments are currently approved and what would be the advantages of the method proposed in this manuscript compared to those currently used?
13- It is also necessary to make it clear that the proposed method can be efficient in the treatment of disseminated intravascular coagulation, but as for the other intoxication effects that may arise, what to do?
14- It would be interesting for the authors to discuss what are the side effects reported in patients treated with recombinant alpha thrombomodulin, and what to do to avoid accentuating the symptoms presented by injured patients!
15- It also needs to be mentioned that the authors used only one concentration of alpha thrombomodulin in the experiments carried out throughout the work. It would be interesting for them to have carried out experiments using varying concentrations of alpha thrombomodulin, which could bring important additional data to conclusions about the treatment of injured people.
16- Finally, it would be interesting for the authors to show some laboratory parameters of renal function markers, for example urea and creatinine, in the animals studied and treated with venom and alpha thrombomodulin, since one of the causes of death in patients is renal failure! Was this thought or done?
17- About discussion. Between lines 167 and 170 the authors wrote ….This clearly showed that rTM does not act directly on R.tigrinus venom. Moreover, as no experiments have been conducted with the addition of rTM inhibitors, the mechanism by which the activity of R. tigrinus venom is canceled has not yet been elucidated in the in vitro experiments as conducted in this study. …. From these observations it is concluded that the toxins of the venom are still active to act in other structures of the tissues of the victims. This raises concerns for the physicians involved, as other harmful activities may appear, irrespective of the clotting effects. What do the authors say about this?
18- Between lines 176 and 177 the authors wrote …. However, even when the same amount of rTM was administered 2 h after the administration of venom, no life-saving effect was observed (Figures 2 and 3). These data need to be further discussed in the text, since, if this proposed treatment is accepted by the medical community, this procedure, as in the case of serum therapy also, needs to be started as soon as possible to obtain therapeutic efficacy.
19- Between lines 187 and 190 the authors wrote …. The difference in the effects of rTM and R. tigrinus antivenom in the in vivo experimental system appeared in the amount of D-dimer as a blood coagulation marker. Antivenom had little effect on the appearance of D-dimer, but rTM significantly suppressed the appearance of D-dimer [23]. It was speculated that the difference between the two was due to the difference in the mechanism of action. ….. Here the authors missed a great opportunity to make comparative analysis of the two treatment methods, performing some additional experiments and using their model, and from there, they would surely obtain a lot of information that would enrich this publication.
20- Please write through out of the text in italic all the terms as Rhabdophis tigrinus, in vitro and in vivo.
Author Response
Dear reviewer 2,
Apr. 29st, 2022
Thank you for your comment on the draft of Toxins-1679860.
Below is a response to each of the three reviewers who received many useful comments.
We will attach the toxins-1679860 (2) .pdf file as a revised paper.
To reviewer 2
General Comments
In this manuscript the authors are describing their experimental data on the use of the recombinant protein Alpha Thrombomodulin, as a possible drug in the treatment of accidents with the Asian snake Rhabdophis tigrinus, seeking to inhibit complications of Disseminated Intravascular Coagulation. The authors studied the protective effects of treatment with Alpha Thrombomodulin in rats exposed to the venom experimentally and treated with recombinant Alpha Thrombomodulin after this exposure. Throughout the text, the authors report their findings with emphasis on the normalization of coagulation parameters in the blood of animals and from there they report the possibility of using this recombinant protein in the treatment of injured patients. The text is well-written, easy to understand, and theoretically well-founded. However, there are suggestions that authors can incorporate into a revised version to increase reader interest and improve some definitions.
Thank you for understanding the purpose of the dissertation. We responded to the comments pointed out as follows.
Comments
- 1. Please, as written on line 38, substitute throughout of the text the words molecular weight for molecular mass. I know that several published texts bring this term, but it is incorrect! Molecules have no weight! Rice, meat have weight... Molecules have molecular masses. We don't have weight spectrometry but instead mass spectrometry....
We have corrected it as you pointed out.
- 2. Between lines 42 to 44 the authors wrote.... The venom fractions also exhibited specific proteinase activity on human fibrinogen (FIB), but the substrates for matrix metalloproteinase, such as collagen and laminin, were not hydrolyzed [6]. I ask … are there data on the hydrolysis of plasma and vascular extracellular matrix proteins such as Fibronectin, von Willebrand Factor and Vitronectin, which are Extracellular Matrix molecules involved in platelet adhesion and aggregation?
Unfortunately, data on the hydrolysis of plasma and vascular extracellular matrix proteins such as fibronectin, von Willebrand factor, and vitronectin, which are extracellular matrix molecules involved in platelet adhesion and aggregation by R. tigrinus venom, have not been investigated.
- 3. Between lines 50 and 51 the authors wrote...Currently, an unapproved drug, tigrinus antivenom, is used as a therapeutic agent for R. tigrinus envenomation, but there are restrictions on its use… What would these restrictions be, in addition to those already mentioned in the difficulty of access and non-approved drugs. Would the serum be a causative agent of severe allergenic responses? It would be good for the authors to discuss more details here about this! Indicating possible advantages of the proposed alternative!
The following sentences have been added as explanations for the points pointed out.
“R.tigrinus antivenom is a horse plasma-derived preparation immunized with R. tigrinus venom, and when administered to humans, there is a risk of causing adverse reactions such as serum sickness and anaphylaxis due to horse-derived protein”.
- 4. Between lines 51 and 55 the authors wrote...There are also several approved drugs already on the market for the treatment of DIC [12–14]. Among those drugs, there is no record of DIC treatment with venomous snakes, but recombinant alpha thrombomodulin (rTM) is often used as a drug that is always available in clinical practice [15]. …Although the authors have not carried out comparative analyzes between drugs already used in the treatment of DIC and recombinant alpha thrombomodulin, here, it would be interesting for them to comment on possible advantages of the proposed method and, in the future, make these comparisons in pre-clinical or clinical trials!
The following sentences have been added as explanations for the points pointed out.
“None of the approved DIC treatments listed here have been used to treat snake ven-om-derived DIC. Therefore, in the future, we plan to investigate the therapeutic effect of snake venom on DIC.”
- 5. Between lines 70 and 71 the authors wrote...In this study, we confirmed the effect of rTM on tigrinus venom in an in vitro blood coagulation system as a therapeutic agent for the DIC state of R. tigrinus bites. …It would be better to write putative therapeutic agent since the method is being tested and its initial pre-clinical phase!
As pointed out, we made it an expression that adds “putative” to “therapeutic agent” as an expression of the part at the end of the introduction.
- 6. Although Figure 1 shows robust data on the inhibition of blood clotting induced by tigrinus venom at different concentrations, caused by the presence of alpha thrombomodulin in the in vitro system using reference plasma. I missed more direct graphs of Prothrombin Times, Partial Activated Prothrombin Time, Thrombin Time, Platelet Count, Platelet Aggregation, Clotting Time, Clot Retraction, among other laboratory parameters used for blood clotting analysis!
Other test parameters used in the blood coagulation analysis pointed out are not measured in vitro such as prothrombin time, partially activated prothrombin time, thrombin time, and platelet count.
- 7. Between lines 100 and 101 the authors wrote …In the in vitro measurement system shown in Figure 1, the activity of tigrinus venom was canceled by rTM. In my opinion, in a revised text It would be better to write that the coagulation activities of the venom were inhibited, because others activities cannot have been inhibited.
Replaced the text "the activity of R. tigrinus venom was canceled by rTM" with "the coagulation activities of the R. tigrinus venom were inhibited by rTM, because others activities cannot have been inhibited."
- 8. What does this horizontal line in figure 1 mean in the value of 1.3 cloting time? There are no comments in the figure legend!
I'm sorry that there is no explanation about this horizon in legends. Other reviewers have pointed out Figure 1, so we rewrote this figure. A standard deviation was added to each measurement point for the coagulation time of the standard serum with R. tigrinus venom and rTM, and the two correlation curves were compared by the parallel line assay method to quantify the suppression of coagulation by R. tigrinus venom by rTM. we added this explanation to 2.1 to make it easier to understand.
- 9. Between lines 107 and 108, discussing about figure 2, the authors wrote …However, the administration of rTM 0.5, 0.33, and 0.17 h after venom administration saved 33.3%, 100%, and 66.7% of the rats, respectively. How do the authors explain that for a shorter time of administration of alpha thrombomodulin (0.17h), the protection was inferior to the time of 0.33h? Would it be individual variation of the animals? Also because only 3 animals were used and this number of animals could be insufficient for such an analysis. Sounds like a small population to me to assess an important postulated clinical effect!
The therapeutic effect of rTM after administration of R. tigrinus pointed out differs depending on the administration time of 0.17,0.33,0.5 hours because of the variation in the response in the experiment with 3 rats in 1 group. We also consider it to be the most applicable explanation. Therefore, we think that more constant experimental results can be obtained by conducting experiments with an increased number of rats in one group. We will reflect this point in our consideration.
- 10. Between lines 108 to 113, discussing about figure 2, the authors wrote …When the significant difference between the two groups was calculated using a statistical method for the life-saving effect shown in Figure 2 between … In my opinion, the authors must indicate the statistical method used
As a correction of the point pointed out, the expression was corrected that the two groups were compared using "Fisher's exact test" instead of "statistical method".
- 11. On the graphs shown in Figure 3, there is a lot of overlap, making it difficult to interpret the results. Did the authors not think of presenting the data in the form of histograms, which could make interpretation easier? Alternatively they could do the graph with different colors which would make the data interpretation easier too!
As you pointed out, the display of Figure 3 is difficult to understand, so we have revised the expression to color-code each group.
- 12. The authors already discuss in the abstract that although serum therapy is an efficient treatment for victims of Rhabdophis tigrinus snake bites, this treatment is difficult to use due to the difficulty in obtaining the serum and also because the serum is not officially approved by the regulatory agencies in Japan. Would it be interesting for them to show which treatments are currently approved and what would be the advantages of the method proposed in this manuscript compared to those currently used?
The current treatment method with R. tigrinus antivenom is that antivenom is 1) difficult to obtain because it is an unapproved drug, and 2) because it is a horse plasma-derived preparation, there is a risk of causing adverse reactions such as serum sickness and anaphylaxis when administered to patients. The advantages of rTM, which we have shown in this study as a therapeutic drug, are that it is an approved drug and is easily available and that there are few side reactions.
- 13. It is also necessary to make it clear that the proposed method can be efficient in the treatment of disseminated intravascular coagulation, but as for the other intoxication effects that may arise, what to do?
We understand that the indications of intoxications mean the treatment of toxicity other than DIC by R. tigrinus venom. R. tigrinus venom also has hemolytic activity. The most problematic toxic effect of R. tigrinus bites is the condition of DIC, which causes the blood coagulation system to fail, causing cerebral hemorrhage and death. The primary purpose of this study is treatment aimed at improving the DIC situation, which is directly linked to death.
- 14. It would be interesting for the authors to discuss what are the side effects reported in patients treated with recombinant alpha thrombomodulin, and what to do to avoid accentuating the symptoms presented by injured patients!
The side effects of rTM pointed out are described in the discussion.
- 15. It also needs to be mentioned that the authors used only one concentration of alpha thrombomodulin in the experiments carried out throughout the work. It would be interesting for them to have carried out experiments using varying concentrations of alpha thrombomodulin, which could bring important additional data to conclusions about the treatment of injured people.
We think it is very meaningful to plan an experiment to administer multiple concentrations of rTM to rats treated with R. tigrinus venom. We would definitely like to plan for the next experimental plan.
- 16. Finally, it would be interesting for the authors to show some laboratory parameters of renal function markers, for example urea and creatinine, in the animals studied and treated with venom and alpha thrombomodulin, since one of the causes of death in patients is renal failure! Was this thought or done?
This is a very suggestive point. In this experiment, we did not measure the values related to renal function in rats as proposed. In future experiments, we aim to measure renal function markers for analysis of rat pathology.
- 17. About discussion. Between lines 167 and 170 the authors wrote …This clearly showed that rTM does not act directly on tigrinus venom. Moreover, as no experiments have been conducted with the addition of rTM inhibitors, the mechanism by which the activity of R. tigrinus venom is canceled has not yet been elucidated in the in vitro experiments as conducted in this study. …. From these observations it is concluded that the toxins of the venom are still active to act in other structures of the tissues of the victims. This raises concerns for the physicians involved, as other harmful activities may appear, irrespective of the clotting effects. What do the authors say about this?
Since the pointed out rTM does not directly neutralize the R. tigrinus venom, the precautions for clinical use are described in the discussion.
- 18. Between lines 176 and 177 the authors wrote …. However, even when the same amount of rTM was administered 2 h after the administration of venom, no life-saving effect was observed (Figures 2 and 3). These data need to be further discussed in the text, since, if this proposed treatment is accepted by the medical community, this procedure, as in the case of serum therapy also, needs to be started as soon as possible to obtain therapeutic efficacy.
Since there is no effect unless the pointed out rTM is administered at an early stage of R. tigrinus bite, points to note regarding clinical use are described in the discussion.
- 19. Between lines 187 and 190 the authors wrote …. The difference in the effects of rTM and tigrinus antivenom in the in vivo experimental system appeared in the amount of D-dimer as a blood coagulation marker. Antivenom had little effect on the appearance of D-dimer, but rTM significantly suppressed the appearance of D-dimer [23]. It was speculated that the difference between the two was due to the difference in the mechanism of action. …Here the authors missed a great opportunity to make comparative analysis of the two treatment methods, performing some additional experiments and using their model, and from there, they would surely obtain a lot of information that would enrich this publication.
We think that is exactly what you pointed out. Unfortunately, there is no further discussion in this part about experiments that suppress the increase in D-dimer with early administration of rTM, as we have not conducted additional experiments related to its mechanism of action. However, in the paragraph two below the pointed out part, we have added a discussion of the mechanism of action of DIC therapeutic effects other than rTM snake venom.
- 20. Please write through out of the text in italic all the terms as Rhabdophis tigrinus, in vitro and in vivo.
The words Rhabdophis tigrinus, in vitro and in vivo that you pointed out have been made into italics.

Reviewer 3 Report
The authors present an investigation of the effects of recombinant thrombomodulin alfa (rTM) on the effect of venom from Rhabdophis tigrinus on coagulation in vitro and in vivo. They contend that rTM is protective against the coagulopathic effects of the venom. The work has major issues as subsequently described.
Results.
2.1. The amounts of venom and rTM are given in microgram quantities without volume of reaction present except in the x-axis of figure 1. What kind of human plasma was used – I see that it is commercially obtained, but is it citrated and kept frozen at -80 degrees C? Is it lyophilized? What are the activities of the coagulation enzymes and fibrinogen? All of this needs to be presented. The rationale for any of these concentrations is not provided, and the representation of the mixture is not standard. Provide actual micrograms/ml of these mixtures, or in the case of rTM, U/ml. Why was there no statistical analyses between the slopes of the two experimental conditions across concentration of venom? Why is standard deviation not presented? In summary, there is no link of these in vitro results with in vivo concentrations.
2.2. What is the rationale for the dose of 1 mg/kg of rTM? Where is the concentration-response relationship? What is the dose of venom in micrograms/kg? The authors only used 3 rats for 5 groups, which cannot provide adequate statistical power. What was the power observed post hoc? The significance of the observed results was P=0.05 when the statistics section stated that P<0.05 was significant. The authors also need to represent the data in a clearer manner, as the symbols and lines of the groups overlap with two of the groups almost indistinguishable. This series of experiments are not adequate to address the issue of survival.
Were the animals subjected to the experiment immediately after anesthesia and surgery to place intravascular catheters? Was there a period of recovery? The authors do not provide these details, and I wonder if mortality would be decreased if the animals were allowed to recover for a day.
2.3. This section is confusing. Are the three rats presented the same rats as in 2.2? Were the other rats that died sampled as well, but without their data being presented? The authors first note that the blood values presented in Figure 3 are from the rats that all lived. However, they then note rats from the other groups that did not die. Some of the groups have standard deviation and some do not, presumably as only one rat remained alive. The weight of the animals are not presented, and the removal of 1.5 ml of blood over time with replaced with an equal amount of saline results in one third of the intravascular volume before sampling. What percentage of circulating estimated blood volume was removed by the authors? Statistical significance is indicated, but it is not clear what the differences are between – groups, time, etc.? The method used to determine significance is also unknown based on the statistical section – one-way ANOVA over time with what post hoc test? How can there be significance if there is only one data point compared to three data points in another group? The time points and time span of observation are also different between all 4 panels.
In conclusion, this investigation provides data that is not justified based on concentration-response data for rTM, is statistically underpowered secondary to insufficient animals per group, not properly analyzed statistically, and physiologically not characterized to the point of repeatability.
Author Response
Dear reviewer 3,
Apr. 29st, 2022
Thank you for your comment on the draft of Toxins-1679860.
Below is a response to each of the three reviewers who received many useful comments.
We will attach the toxins-1679860 (2) .pdf file as a revised paper.
To reviewer 3
The authors present an investigation of the effects of recombinant thrombomodulin alfa (rTM) on the effect of venom from Rhabdophis tigrinus on coagulation in vitro and in vivo. They contend that rTM is protective against the coagulopathic effects of the venom. The work has major issues as subsequently described.
We have made the following corrections to the points that were pointed out regarding the results. We hope that this amendment will improve the problems in this paper.
Results.
2.1. The amounts of venom and rTM are given in microgram quantities without volume of reaction present except in the x-axis of figure 1. What kind of human plasma was used – I see that it is commercially obtained, but is it citrated and kept frozen at -80 degrees C? Is it lyophilized? What are the activities of the coagulation enzymes and fibrinogen? All of this needs to be presented. The rationale for any of these concentrations is not provided, and the representation of the mixture is not standard. Provide actual micrograms/ml of these mixtures, or in the case of rTM, U/ml. Why was there no statistical analyses between the slopes of the two experimental conditions across concentration of venom? Why is standard deviation not presented? In summary, there is no link of these in vitro results with in vivo concentrations.
For human plasma that was pointed out, the components of the lot used, the manufacturing method, and the activity value of rTM are additionally described in 5.1.
In addition, Figure 1 was rewritten to show the average value and standard deviation of the coagulation time at each concentration of R. tigrinus venom, and statistical analysis of the slope when rTM was added and statistical comparison of the two experimental conditions were performed and added to 2.1.
2.2. The comments for Result 2.2 are divided into the following nine, and the correspondence to each is shown.
1) What is the rationale for the dose of 1 mg/kg of rTM?
2) Where is the concentration response relationship?
3) What is the dose of venom in micrograms/kg?
4) The authors only used 3 rats for 5 groups, which cannot provide adequate statistical power.
5) What was the power observed post hoc?
6) The significance of the observed results was P=0.05 when the statistics section stated that P<0.05 was significant.
7) The authors also need to represent the data in a clearer manner, as the symbols and lines of the groups overlap with two of the groups almost indistinguishable.
8) This series of experiments are not adequate to address the issue of survival. 
9) Were the animals subjected to the experiment immediately after anesthesia and surgery to place intravascular catheters? Was there a period of recovery? The authors do not provide these details, and I wonder if mortality would be decreased if the animals were allowed to recover for a day.
Below are the answers to the comments made in Result 2.2.
1) This experiment was the first dose of snake venom to a rat DIC model, and in planning it, the response to rTM in rats was weaker than in humans, so it was 100 times (1 mg / kg) the clinical dose in humans (10 ug / kg) was administered.
2) As we answered in 1), we did not go with other concentrations. Therefore, it is unclear about the concentration reaction.
3)  The dosage in rats with toxicity recovered and lyophilized by the method described in 5.3 of Materials and Methods is shown. This dose is for creating the DIC model presented in reference 23.
4)  This point is as pointed out. Therefore, as shown in Figure 2, the therapeutic effect of rTM administered within 0.5 hours after venom administration does not depend on the administration time. This experimental data shows the experimental results of an attempt to see if rTM, one of the anti-DIC drugs, is effective for the rat DIC model. Therefore, we believe that we will plan experiments in which the dose and administration time of rTM are changed using more rats in the future, and definitive results will be obtained.
5)  We don't understand what “the power observed post hoc” really means.
6) As you pointed out. The notation of P = 0.05 was a clerical error of p <0.05. We will correct it to p <0.05.
7) In order to eliminate the incomprehensibleness of Figure 2 that was pointed out, each group is color-coded to show Figure 2.
8) Although the point pointed out cannot be denied, at least rats can be treated with rTM within 0.5 hours after venom administration in a rat DIC model that dies 100% within 48 hours after administration of rTM 2 hours after administration of venom alone or venom. We have been saved. In total, 6 out of 9 animals in 3 groups have been saved, so it is considered that at least a certain therapeutic effect by rTM is recognized.
9) It is the author's mistake that there was no mention of the recovery period after the rat was cannulated. Add to 5.2 of Materials and Methods that 48 hours was specified as the recovery period after treatment for rats. It is the author's mistake that there was no mention of the recovery period after the rat was cannulated. Add to 5.2 of Materials and Methods that 48 hours was specified as the recovery period after treatment for rats.
2.3. The comments for Result 2.3 are divided into the following five, and the correspondence to each is shown.
1)This section is confusing.
2)Are the three rats presented the same rats as in 2.2?
3)Were the other rats that died sampled as well, but without their data being presented? The authors first note that the blood values presented in Figure 3 are from the rats that all lived. However, they then note rats from the other groups that did not die. Some of the groups have standard deviation and some do not, presumably as only one rat remained alive.
4)The weight of the animals are not presented, and the removal of 1.5 ml of blood over time with replaced with an equal amount of saline results in one third of the intravascular volume before sampling. What percentage of circulating estimated blood volume was removed by the authors?
5)Statistical significance is indicated, but it is not clear what the differences are between – groups, time, etc.? The method used to determine significance is also unknown based on the statistical section – one-way ANOVA over time with what post hoc test? How 2 can there be significance if there is only one data point compared to three data points in another group? The time points and time span of observation are also different between all 4 panels.
1)There was an error in the notation. Corrected the error.
2)Rats are rats showing data in Results 2.2
3)The statement that only surviving rats were shown is incorrect. To be precise, all dead rats that could be voted during life were shown in Figure 3 by measuring blood coagulation markers. Corrected an error.
4)The weight of the rat used in the experiment is described in 5.2 of Materials and Methods.
5)The comparison between the groups shown in Figure 3 is a one-to-one comparison between the toxin-only group and the rTM-treated group, so the two-way arrangement method among the statistical methods was used for comparison. As a result, the group showing p <0.01 is shown by **.
In conclusion, this investigation provides data that is not justified based on concentration response data for rTM, is statistically underpowered secondary to insufficient animals per group, not properly analyzed statistically, and physiologically not characterized to the point of repeatability.
As mentioned in the answers to the comments in 2.1, 2.1, 2.3, this study is the result of investigating the therapeutic effect of rTM on in vitro and in vivo experimental systems against R. tigrinus venom, so there are some deficiencies pointed out. However, we would like to announce it as a preliminary result before the full-scale examination.

Round 2
Reviewer 3 Report
The text and graphics have dramatically improved. Well done.
I accept that this is a preliminary work. The authors really need to learn about statistics and what statistical power means. The work has far too few animals, but I will let this go by in the hopes of a more extensive and definitive work.
I found only one minor mistake.
Line 304. “After placement of these two catheters, the rats were bred for 48 hours…” Do the authors mean bled, not bred?
Author Response
Dear reviewer 3,
Apr. 30st, 2022
Akihiko Yamamoto, DVD, Ph. D.
National Institute of Infectious Diseases
Thank you for your comment on the draft of Toxins-1679860.
Below is a response to the reviewer who received many useful comments.
The text and graphics have dramatically improved. Well done.
I accept that this is a preliminary work. The authors really need to learn about statistics and what statistical power means. The work has far too few animals, but I will let this go by in the hopes of a more extensive and definitive work.
I found only one minor mistake.
Line 304. “After placement of these two catheters, the rats were bred for 48 hours…” Do the authors mean bled, not bred?
In response to your harsh and warm comments, we managed to revise the paper to deserve a presentation at Toxins. We are really thankful to you.
As for the minor point you pointed out, the meaning of this sentence is certainly not breeding but blood sampling. The text you pointed out has been corrected as follows so as not to give a misunderstanding.
“After placing these two catheters, the rats were subjected to the experiment after a 48-hour post-treatment recovery period.”
